# Correlation between Executive Network Integrity and Sarcopenia in Patients with Parkinson’s Disease

**DOI:** 10.3390/ijerph16244884

**Published:** 2019-12-04

**Authors:** Chih-Ying Lee, Hsiu-Ling Chen, Pei-Chin Chen, Yueh-Sheng Chen, Pi-Ling Chiang, Cheng-Kang Wang, Cheng-Hsien Lu, Meng-Hsiang Chen, Kun-Hsien Chou, Yu-Chi Huang, Wei-Che Lin

**Affiliations:** 1Department of Diagnostic Radiology, Kaohsiung Chang Gung Memorial Hospital and Chang Gung University College of Medicine, Kaohsiung 83305, Taiwan; rose80113@hotmail.com (C.-Y.L.); suring.tw@gmail.com (H.-L.C.); pachin1117@gmail.com (P.-C.C.); qqqqqq4545@cgmh.org.tw (Y.-S.C.); lovage@cgmh.org.tw (P.-L.C.); moonrbt@gmail.com (C.-K.W.); sperfect1101@gmail.com (M.-H.C.); 2Department of Neurology, Kaohsiung Chang Gung Memorial Hospital and Chang Gung University College of Medicine, Kaohsiung 83305, Taiwan; chlu99@ms44.url.com.tw; 3Brain Research Center, National Yang-Ming University, Taipei 11221, Taiwan; dargonchow@gmail.com; 4Institute of Neuroscience, National Yang Ming University, Taipei 11221, Taiwan; 5Department of Physical Medicine and Rehabilitation, Kaohsiung Chang Gung Memorial Hospital and Chang Gung University College of Medicine, Kaohsiung 83305, Taiwan; hyuchi@gmail.com

**Keywords:** neurodegenerative disorder, muscle atrophy, executive function, fronto-striato-thalamic circuits, diffusion tensor imaging

## Abstract

*Background*: Sarcopenia is critically associated with morbidity and mortality in the progression of Parkinson’s disease (PD). However, analyses of clinical severity and brain changes, such as white matter (WM) alterations in PD patients with sarcopenia are limited. Further understanding of the factors associated with sarcopenia may provide a focused screen and potential for early intervention in PD patients. *Methods*: 52 PD patients and 19 healthy participants accepted dual-energy X-ray absorptiometry to measure the body composition. Using diffusion tensor imaging, the difference of WM integrity was measured between PD patients with sarcopenia (PDSa) and without sarcopenia (PDNSa). Multivariate analysis was performed to explore the relationships between clinical factors, WM integrity, and sarcopenia in PD patients. *Results*: 21 PD patients (40.4%) had sarcopenia. PDSa had a higher Unified Parkinson’s Disease Rating Scale (UPDRS III) score, lower body mass index (BMI) and lower fat weight compared with the PDNSa. Additionally, PDSa patients exhibited lower fractional anisotropy accompanied by higher radial diffusivity and/or higher mean diffusivity in the fronto-striato-thalamic circuits, including bilateral cingulum, left superior longitudinal fasciculus, left genu of corpus callosum, and right anterior thalamic radiation, which participate in the executive function. In addition, decreased muscle mass was associated with worse WM integrity in these regions. Multiple linear regression analysis revealed that WM integrity in the left cingulum, right anterior thalamic radiation, together with gender (male) significantly predicted muscle mass in PD patients. *Conclusions*: WM alterations in the executive network, such as the fronto-striato-thalamic circuits, may indicate a risk factor for ongoing sarcopenia in PD patients. The effectiveness of using executive function to serve as a prodromal marker of sarcopenia in PD patients should be evaluated in future studies.

## 1. Introduction

Parkinson’s disease (PD) is a neurodegenerative movement disorder, which presents with motor and non-motor symptoms, such as cognitive declined and physical frailty, declining clinical conditions and quality of life [1]. A high prevalence of sarcopenia in PD patients with qualitative and quantitative deterioration of skeletal muscle has been reported [2]. In particular, patients presenting advanced age, longer disease duration, and higher disease severity demonstrate susceptibility to developing sarcopenia [2]. Commonly occurring features, such as motor impairment, difficulty in swallowing and malnutrition, are risk factors that may accelerate the progression of sarcopenia [3]. While instances of concurrent diagnoses of sarcopenia and PD are potentially high, more thorough investigation is required. Of note, the associations between sarcopenia and neurological diseases remain unclear, such as the specific pathways shared by PD and sarcopenia, and their interactions. 

The pathology of Parkinson’s disease involves cerebral white matter (WM), while several affected regions may be associated with executive function [4]. Executive dysfunction is noted at the early stage of PD and as a predictor of the development of dementia and frailty [5,6]. Although clinical cognitive impairment and sarcopenia may concurrently present in PD patients, it is unclear whether the specific pathological changes of brain structure in PD patients, such as executive function-related WM alterations, are associated with onset of sarcopenia. Thus, the impact of executive dysfunction and other related factors are critical issues requiring further study in PD patients with sarcopenia.

Elucidation of the factors associated with sarcopenia could provide a focused screen and possibility for early intervention in PD patients. We hypothesize that brain damage causing cognitive impairment, such as executive dysfunction, may predict the presence of sarcopenia. This study aims to (1) measure muscle and fat in PD patients by dual-energy X-ray absorptiometry; (2) evaluate WM integrity in PD patients using diffusion tensor imaging (DTI); (3) evaluate the relationships between sarcopenia and WM alterations in PD patients.

## 2. Materials and Methods 

### 2.1. Participants

The 52 PD patients included in the study (21 men and 31 women; mean age 61.7 ± 10.6 years) had no other neurological diseases, psychiatric illnesses, contraindication to magnetic resonance imagining (MRI), nor psychotropic medication. The patients with idiopathic PD were diagnosed by an experienced neurology specialist in the Neurology Department of Kaohsiung Chang Gung Memorial Hospital, in accordance with the United Kingdom Brain Bank criteria. The mean duration of disease was 2.2 ± 2.2 years. The functional status and disease severity were assessed with the Unified Parkinson’s Disease Rating Scale (UPDRS), the modified Hoehn and Yahr Staging Scale, and the Schwab and England Activities of Daily Living Scale.

All the PD patients had dopaminergic medications. The mean levodopa equivalent daily dose (LEDD) [7] was 505.14 ± 376.43 mg, and the mean duration of treatment was 20.8 ± 25.0 months. For comparison, 19 healthy participants (10 men and 9 women; mean age 60.3 ± 7.6 years) without sarcopenia, neurological diseases, psychiatric illnesses, head injury, or alcohol/substance abuse were recruited as a normal control group (NC).

This study was approved by the Kaohsiung Chang Gung Memorial Hospital Ethics Committee in Taiwan, and had been conducted in accordance with the Declaration of Helsinki. All the participants provided written informed consent prior to participation in the study. The ethical code of this study is “103-6906A3”.

### 2.2. Sarcopenia-Associated Measurements

The body composition was measured using dual-energy X-ray absorptiometry (Hologic Horizon W software ver.13.6.0.3; Hologic Inc., Marlborough, MA, USA). We calculated the appendicular skeletal muscle mass (ASM) by summing muscle mass in the extremities. The appendicular skeletal muscle mass index (ASMI) was defined as ASM in kilograms divided by the square of the height in meters. According to the Sarcopenia and Translational Aging Research in Taiwan [8], sarcopenia was defined as ASMI(ASM/h^2^) <6.76 for men and <5.28 for women. The PD patients were then divided into the group with sarcopenia (PDSa) and the group without sarcopenia (PDNSa).

### 2.3. Diffusion Tensor Imaging (DTI) Acquisition 

All subjects were scanned on a 3T MRI system with an 8-channel head coil (General Electric Healthcare, Milwaukee, WI, USA). Using a single shot spin-echo echoplanar imaging sequence, DTI was scanned in the axial plane along the anterior-posterior commissure line (55 slices without gaping, voxel size 1 × 1 × 2.5 mm^3,^ field of view 256 mm, matrix size 256 × 256, number of excitations 3, repetition time/echo time 15,800/77 ms). The diffusion images gradient encoding schemes consisted of 13 non-collinear directions (*b*-value = 1000 s/mm^2^) and one non–diffusion-weighted image (*b*-value = 0 s/mm^2^).

### 2.4. Data Preprocessing

A description of DTI data processing was presented in our previous study [9] as Appendix A. The diffusion tensor model was fitted in each voxel for the fractional anisotropy (FA), mean diffusivity (MD), axial diffusivity (AD), and radial diffusivity (RD) values.

### 2.5. Analysis of Demographic Data Differences between Groups

The demographic data, including age, ASMI, body mass index (BMI), height and Mini-Mental State Examination were compared using the analysis of variance. Gender was compared using the Pearson Chi-square test. The disease duration, UPDRS, the modified Hoehn and Yahr Staging Scale, the Schwab and England Activities of Daily Living Scale, LEDD, and treatment duration were compared using the *t*-test for independent samples. Statistical significance was set at *p* < 0.05. 

### 2.6. Analysis of Body Composition Differences between Groups

The analysis of variance was used to compared the body composition of the participants between three groups. Multiple comparison analysis testing with Bonferroni correction was used to show the group differences.

### 2.7. Analysis of Group Comparison of Fractional Anisotropy (FA) Values

We used the Statistical Parametric Mapping 8 (University College London, UK), which used MATLAB (Mathworks, USA) for voxel-wise group comparisons. We analyzed smoothed, normalized FA images within the framework of a general linear model, whereas analysis of covariance was performed with age and sex as covariates to measure the FA differences between PDSa and PDNSa groups. The FA threshold of the mean WM was set at 0.2 to successfully exclude voxels, which consisted of gray matter or cerebrospinal fluid in most subjects. The statistical inferences were considered significant under the criteria of cluster level family-wise error corrected *p*-value <0.05, with a cluster size of ≥215 voxels, based on the Monte Carlo simulation (3dClusterSim with the following parameters: single voxel *p*-value <0.001, full width at half maximum = 8 mm with gray matter mask, and 5000 simulations). The most probable fiber tracts and anatomic locations of each significant cluster were determined using the FSL atlas tool (http://www.fmrib.ox.ac.uk/fsl/fslwiki/Atlases).

### 2.8. Analysis of Regions of Interest

Based on whole-brain voxel-wise comparisons, the regions of interest (ROI) were determined the mean FA value of each significantly different area between PDSa and PDNSa. We used the Marsbar toolbox (http://marsbar.sourceforge.net/download.html) to extract the ROI masks. Multivariate analysis of covariance was used to compared the mean DTI-related indices of these areas between groups, controlling for age, gender, and LEDD. Partial correlation was used to determine the relationship between FA of the ROI and ASMI, controlling for age, gender, and LEDD. Linear regression was used to illustrate the trend of lower ASMI with lower FA values in the ROI. Then, stepwise multiple linear regression was used to predict the association between ASMI and FA of the ROI, with age, gender, disease duration, UPDRS III, treatment duration, and LEDD as covariates. Figure 1 presents the flow chart of the study’s algorithm.

## 3. Results 

### 3.1. Demographic and Clinical Characteristics

The demographic characteristics results of the three groups (NC, PDNSa, and PDSa), and clinical severity of PD were listed in Table 1. Twenty-one patients with PD (40.4%) exhibited sarcopenia. The ASMI and BMI were lowest in the PDSa groups. The UPDRS III and total score were significantly higher in the PDSa than in the PDNSa groups. There were no significant differences in age, gender, height, and Mini-Mental State Examination between the three groups. Also, there were no significant differences in disease duration, UPDRS I and II score, the modified Hoehn and Yahr Staging Scale, the Schwab and England Activities of Daily Living Scale, LEDD, or treatment duration between the PDSa and the PDNSa groups.

### 3.2. Body Composition

The body composition data of the three groups was presented in Table 2. Compared to the NC, the PDNSa groups had higher fat weight and higher fat percentage in extremities. The PDSa groups had the least fat weight of the three groups. The PDNSa groups had lower muscle percentage than the NC. The PDSa groups had lower muscle weight than the NC. These data imply that sarcopenia is not only a disease resulting in decreases of muscle mass but also of fat mass.

### 3.3. Brain Alterations in DTI

Table 3 and Figure 2 presented the ROI with significant differences in FA values between the PDSa and the PDNSa. The PDSa had lower FA values in the left occipital WM, left superior longitudinal fasciculus (SLF), left genu of corpus callosum (CC), right temporal WM, right parahippocampal gyrus, right anterior thalamic radiation (ATR), right anterior cingulate, and bilateral cingulum.

Table 3 also presented the lower FA values in the PDSa group associated with differences in other diffusivity indices, including (1) increased MD and RD values in the left occipital WM, left SLF, right temporal WM, right parahippocampal gyrus, and bilateral cingulum; (2) increased RD values in the left genu of CC, right temporal WM, and right ATR. There were no significant differences in the AD values of these ROI between the two groups.

Using partial correlation, we found significantly associations between ASMI and FA of the ROI in the left SLF, left genu of CC, right ATR, and left cingulum, after controlling for age, gender, and LEDD. Statistical significance was set at *p* < 0.0045 (0.05/11). Figure 3 demonstrated the trend of lower ASMI with lower FA values in those regions using linear regression. FA of right ATR, FA of left cingulum, and gender significantly predicted the skeletal muscle mass in PD patients. The β-coefficient of ASMI associated with FA of right ATR was 0.316 (*p* = 0.012), with FA of left cingulum was 0.320 (*p* = 0.010), and with gender was 0.340 (male, *p* = 0.003). The final model was more informative (R2 = 53.4%, adjusted R2 = 48.2%).

## 4. Discussion

This study reveals that the PDSa group has a higher UPDRS III score and lower BMI compared with the PDNSa group. DTI reveals the WM integrity alterations in the PDSa as lower FA accompanied with higher RD and/or higher MD mainly in the fronto-striato-thalamic circuits, including the bilateral cingulum, left SLF, left genu of CC, right ATR, and right anterior cingulate, which participate prominently in the executive function. Lower FA values are also noted in the regions of the right parahippocampal gyrus, left occipital and right temporal WM. Moreover, decreased ASMI is associated with reduced FA in the fronto-striato-thalamic circuits. To identify the predictive factor of sarcopenia in PD patients, low FA values in the left cingulum, right ATR, and the gender factor exhibit the strongest correlations with decreased muscle mass. To the best of our knowledge, this is the first study to report the potential pathogenesis of altered WM integrity in PD patients with sarcopenia using DTI.

Consistent with a previous study [10], the PDSa group exhibited a higher UPDRS III score than the PDNSa group, this score is used to evaluate motor skills, including limb movement, gait, and body bradykinesia [11]. Our result indicates that sarcopenia in PD patients is associated with diminished motor abilities. This finding may explain that low body mass index(BMI) [12], decreased lean mass [13], and abnormal ratio of fat mass [12,13] in PD patients notably increase the risk of disability in daily activities [14]. Multiple linear regression analysis demonstrates that the male gender factor is a significant predictor of decreased skeletal muscle mass, consistent with a previous study [2].

Sarcopenia exhibits varying characteristics at different stages of PD [2]. In the early stage of PD, patients more commonly present overweight and central obesity than healthy persons [15], while patients in the late stage commonly present underweight [16]. Our result of the body composition implies that sarcopenia is not only a disease resulting in decreased muscle mass but also fat mass. Our study demonstrates that the PDNSa group exhibits higher fat mass and lower muscle mass than healthy persons, while the PDSa group exhibits the least fat and muscle of the three groups. These findings suggest that sarcopenia in PD patients exhibits a tendency toward higher percentage of fat loss than percentage of muscle loss. Interestingly, this result is similar to a recent study which reveals that fat mass index is reduced in PD patients, and reduced fat is associated with higher risk of motor complications in PD [17], corresponding with the higher UPDRS III score of the PDSa patients in this study. Although the pathophysiology of this finding is unclear, higher BMI, fat mass, and lean body mass in elders have a positive association with better cognitive performance, especially in executive function [18,19]. In contrast, enhanced fat loss resulting from sarcopenia is associated with decreased cognitive function

The fronto-striato-thalamic circuits are five circuits which originate in the prefrontal cortex, project to the striatum, connect to the globus pallidus and substantia nigra, and then connect to thalamic nuclei, finally linking back to the prefrontal cortex [20]. Two of these circuits, including the dorsolateral prefrontal circuit and anterior cingulate circuit, play noticeable roles in executive function [20]. Dopamine depletion in PD patients disrupts the fronto-striato-thalamic circuits, and leads to executive impairment [21]. According to our results, DTI reveals the WM damage to the right anterior cingulate, right ATR, left SLF, and left genu of CC. All of these regions participate, to varying degrees, in the dorsolateral prefrontal circuit or anterior cingulate circuit [22,23,24]. Executive function relies not only on the fronto-striato-thalamic circuits but also on several other regions, such as the medial temporal lobe and parahippocampal cortex [25], which also revealed WM damage in this study. The anterior cingulate circuit is involved in motivation and the reward systems [20]. In PD, the down-regulated dopamine level results in diminished reward signals [26]. Muscle training often occurs in daily activity. Decreased executive function and reward signals result in decreased motivation of movement and declining physical activity [27], thus subsequently leading to disuse muscle atrophy in PD patients [28].

Whether the executive dysfunction and sarcopenia share common etiologies and physiologic alterations, or whether there exists a causal relationship remains unclear. Several possible mechanisms may explain how sarcopenia is related to executive dysfunction. First, executive dysfunction is associated with earlier onset of frailty, which manifestes as a syndrome comprising decreased activity and inadequate dietary intake, possibly triggering excessive muscle wasting [6,27]. Second, adipose tissue secretes leptin, which contributes to prevention of cognitive decline in older adults [29]. Third, a common pathophysiology shared by sarcopenia and executive dysfunction exists in the inflammatory pathways. Elevated circulating inflammatory mediators such as interleukin-6 are detectable in both PD and sarcopenia patients [30,31]. Interleukin-6 is associated with the muscle loss and poor physical performance in PD patients [30], and is inversely associated with executive function and WM volumes [32]. Additionally, body fat mass is an important source of endogenous estrogen after menopause, which may prevent cognitive decline [33].

## 5. Limitations

There indeed several limitations to this study. First, our research contains a relatively small sample size. While sarcopenia is defined using the cutoff points of ASMI provided by the Sarcopenia and Translational Aging Research in Taiwan [8], despite being widely accepted in Taiwan it may not accurately represent the general population of PD patients. More ethnically diverse populations should be considered for future study. Second, ASMI is important for the diagnosis of sarcopenia; however physical performance and muscle strength, such as gait speed and handgrip strength, are also variables used to measure sarcopenia. The absence of these measurements is likely to affect the estimation of sarcopenia. Third, the retrospective case control study approach is not truly appropriate for establishing a causal relationship between sarcopenia and executive dysfunction in PD patients; therefore, a future prospective study is recommended to address this issue. Despite these limitations, we believe this study acts as a foundation for establishing the relationship between WM alterations and development of sarcopenia in PD patients.

## 6. Conclusions

This study demonstrates a high prevalence of sarcopenia in studied PD patients. The results herein suggest that WM alterations in the executive functional network, such as the fronto-striato-thalamic circuits, may indicate a risk factor for ongoing sarcopenia in PD patients. The effectiveness of using executive function as a prodromal marker of sarcopenia in PD patients requires further investigation

## Figures and Tables

**Figure 1 ijerph-16-04884-f001:**
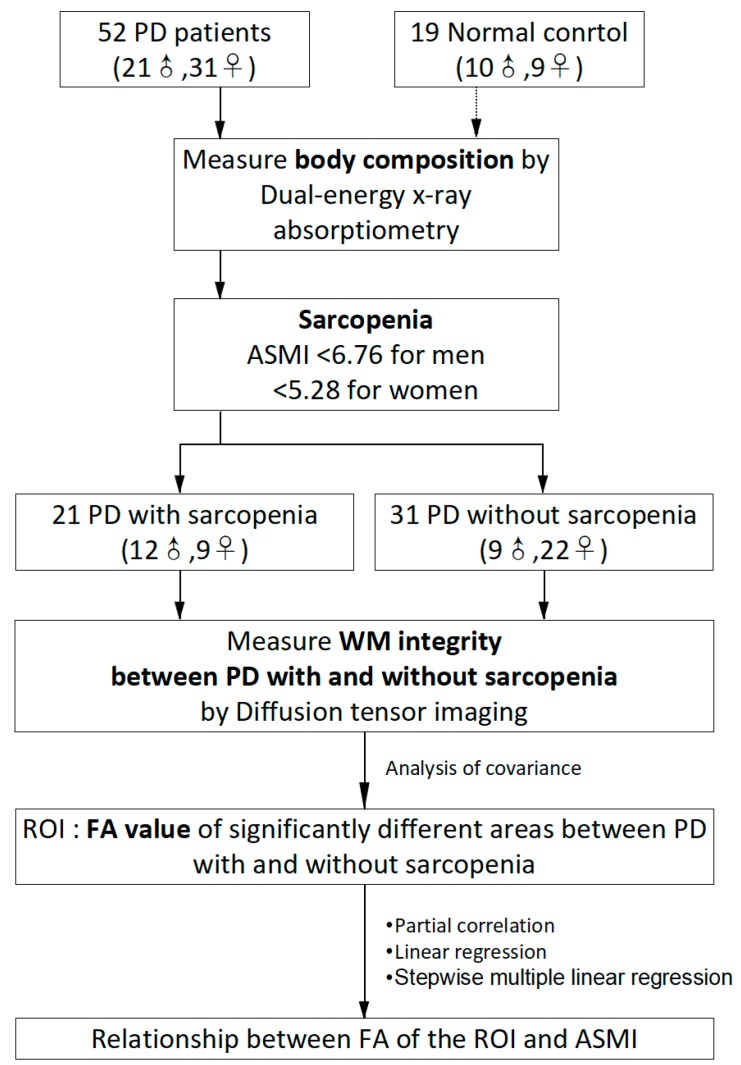
The flow chart of the study’s algorithm. PD = Parkinson’s disease; ASMI = appendicular skeletal muscle mass index; WM = white matter; ROI = regions of interest; FA = fractional anisotropy.

**Figure 2 ijerph-16-04884-f002:**
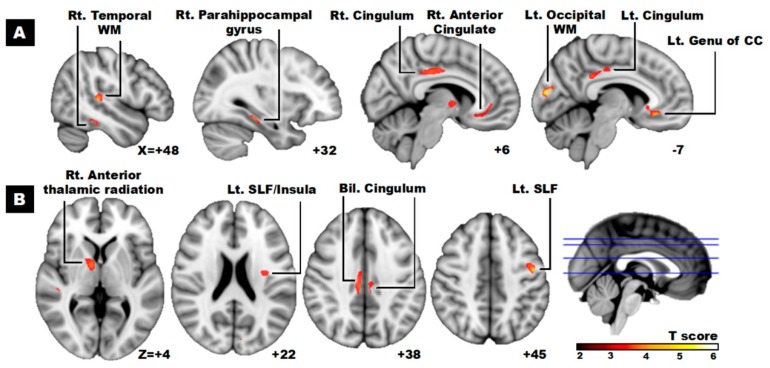
Lower FA values in PD patients with sarcopenia versus without sarcopenia. Anatomical changes are superimposed on the T1 template located in Montreal Neurological Institute (MNI) space in (**A**) sagittal and (**B**) axial view. The numbers in the lower right corner of each image represent the MNI in (A) x- and (B) z-coordinates respectively. WM = white matter; SLF = superior longitudinal fasciculus; CC = corpus callosum.

**Figure 3 ijerph-16-04884-f003:**
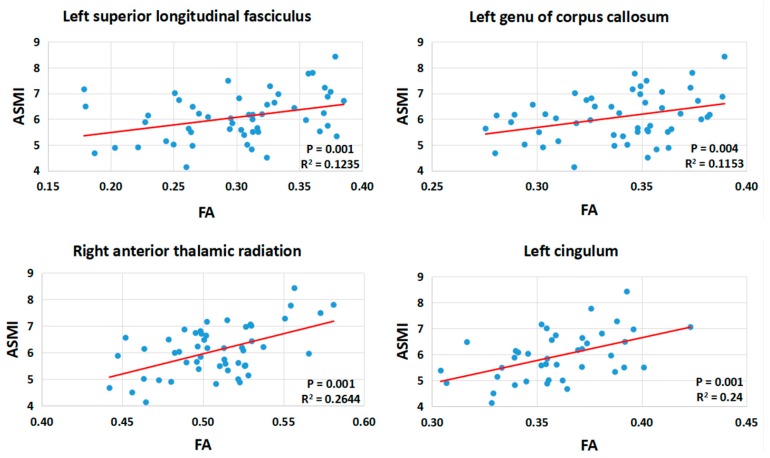
Linear regression between ASMI and FA of the ROI in cluster left superior longitudinal fasciculus, left genu of corpus callosum, right anterior thalamic radiation, and left cingulum. ASMI = appendicular skeletal muscle mass index; FA = fractional anisotropy.

**Table 1 ijerph-16-04884-t001:** Demographic and clinical characteristics of the participants.

	NC (*n* = 19)	PDNSa (*n* = 31)	PDSa (*n* = 21)	*p*
Age (year) ^a^	60.3 ± 7.6	60.3 ± 9.8	63.7 ± 11.6	0.430
Gender (Male) ^b^	10 (52.6%)	9 (29.0%)	12 (57.1%)	0.088
ASMI (ASM/ht^2^) ^a^	7.0 ± 1.1	6.5 ± 0.8	5.5 ± 0.7	<0.001 *
BMI ^a^	25.0 ± 2.6	26.1 ± 3.1	21.4 ± 3.5	<0.001 *
Height(cm) ^a^	163.0 ± 7.5	158.4 ± 7.0	160.1 ± 7.2	0.089
MMSE ^a^	27.0 ± 2.3	26.0 ± 4.6	25.5 ± 4.2	0.519
Disease duration (year) ^c^	―	2.4 ± 2.4	1.9 ± 2.0	0.453
UPDRS ^c^				
I	―	3.4 ± 2.1	3.8 ± 3.0	0.637
II	―	7.2 ± 4.3	9.9 ± 5.4	0.057
III	―	17.6 ± 12.5	29.1 ± 12.4	0.002 *
total	―	28.2 ± 17.0	42.7 ± 17.5	0.005 *
Modified H&Y stage ^c^	―	1.6 ± 1.1	1.6 ± 1.0	0.923
S&E ^c^	―	83.6 ± 16.8	83.5 ± 10.4	0.991
LEDD ^c^	―	439.3 ± 373.6	602.5 ± 367.8	0.126
Treatment duration (month) ^c^	―	21.9 ± 26.5	17.8 ± 22.7	0.568

NC = normal control; PDNSa = Parkinson’s disease without sarcopenia; PDSa = Parkinson’s disease with sarcopenia; MMSE = Mini-Mental State Examination; Modified H&Y stage = Modified Hoehn and Yahr Staging Scale; S&E = Schwab and England Activities of Daily Living Scale; Data are presented as mean ± standard deviation (SD); ^a^ Analysis of variance; ^b^ Pearson Chi square test; ^c^
*t*-test for independent samples; * *p* < 0.05

**Table 2 ijerph-16-04884-t002:** Body composition by dual-energy X-ray absorptiometry of the participants.

	NC (*n* = 19)	PDNSa (*n* = 31)	PDSa (*n* = 21)	*p*
Fat (g)				
Arms	2932.1 ± 907.0	3742.4 ± 1457.9 ^△^	2337.8 ± 802.2 ^△^	<0.001
Legs	6503.6 ± 1885.4	8303.3 ± 3236.6 ^△^	5601.2 ± 1673.6 ^△^	0.001
Four limbs	9435.7 ± 2638.3 *	12,045.7 ± 4571.4 *^△^	7938.9 ± 2422.1 ^△^	0.001
Fat of total ^a^ (%)				
Arms	35.7 ± 10.3 *	43.6 ± 12.0 *^△^	36.9 ± 11.5 ^△^	0.035
Legs	31.4 ± 8.4 *	38.1 ± 10.1 *	33.7 ± 8.5	0.039
Four limbs	32.6 ± 8.7 *	39.7 ± 10.4 *	34.6 ± 9.2	0.035
Muscle mass (g)				
Arms	5093.7 ± 1405.5 ^◆^	4346.0 ± 1366.4	3731.9 ± 984.3 ^◆^	0.006
Legs	13,644.7 ± 2937.4 ^◆^	12,817.8 ± 5261.4	10,350.6 ± 1848.1 ^◆^	0.028
Four limbs	18,738.4 ± 4294.7 ^◆^	17,163.8 ± 6276.4	14,082.5 ± 2781.8 ^◆^	0.015
Muscle mass of total ^a^ (%)				
Arms	60.6 ± 9.8 *	51.3 ± 12.8 *	59.2 ± 11.0	0.011
Legs	65.1 ± 8.3	58.9 ± 9.8	62.8 ± 8.2	0.056
Four limbs	63.8 ± 8.5 *	56.6 ± 10.0 *	61.8 ± 8.6	0.024

^a^ Total = Fat + Bone mineral content + Muscle mass (g); Multiple comparison analysis testing with Bonferroni correction (*p* < 0.05): * NC vs. PDNSa, ^◆^ NC vs. PDSa, ^△^ PDNSa vs. PDSa.

**Table 3 ijerph-16-04884-t003:** Regions showing FA differences among PD patients with and without sarcopenia.

Cluster	MNI Atlas coordinates			Voxel Size	WM Tract	Near Cortical Area	FA, mean ± SD		Tmax	Diffusivity Values (PDNSa−PDSa) (×10^−6^ mm^2^/s)		
	x	y	z				PDNSa	PDSa		MD	AD	RD
1	−5	−89	14	353	Left occipital WM	Left cuneus	0.254 ± 0.048	0.196 ± 0.023	5.82	−131.25 **	−92.18	−150.79 **
2 *	−47	−7	45	394	Left SLF	Left precentral gyrus	0.328 ± 0.045	0.258 ± 0.042	4.81	−131.04 **	−70.45	−161.34 **
3 *	−6	30	−11	409	Left genu of CC	Left anterior cingulate	0.350 ± 0.024	0.321 ± 0.031	4.76	−62.01	−42.95	−71.54 **
4	48	−31	5	459	Right temporal WM	Right superior temporal gyrus	0.383 ± 0.042	0.346 ± 0.037	4.52	−25.31	−13.59	−44.76 **
5	48	−34	−20	227	Right temporal WM	Right fusiform gyrus	0.245 ± 0.051	0.195 ± 0.036	4.23	−63.53 **	−22.84	−83.89 **
6	32	−30	−17	287	Right parahippocampal gyrus	Right parahippocampal gyrus	0.264 ± 0.031	0.222 ± 0.024	4.19	−40.19 **	−9.10	−55.73 **
7 *	10	−2	3	661	Right ATR	Right thalamus	0.520 ± 0.028	0.489 ± 0.028	4.14	−25.84	−3.06	−37.24 **
8	6	−26	36	594	Right cingulum	Right cingulate gyrus	0.373 ± 0.031	0.347 ± 0.042	4.06	−34.91 **	−18.61	−43.05 **
9	4	26	−12	240	Right anterior cingulate	Right anterior cingulate	0.268 ± 0.034	0.219 ± 0.041	4.01	−37.89	−2.19	−57.93
10 *	−7	−39	31	429	Left cingulum	Left cingulate gyrus	0.373 ± 0.026	0.348 ± 0.020	3.96	−24.30 **	−3.12	−34.89 **
11	−32	−10	22	271	Left SLF	Left insula	0.459 ± 0.026	0.424 ± 0.033	3.67	−29.46	−4.67	−41.86 **

Superior longitudinal fasciculus = SLF; Corpus callosum = CC; Anterior thalamic radiation = ATR; FA = fractional anisotropy; MD = mean diffusivity; AD = axial diffusivity; RD = radial diffusivity; MNI = Montreal Neurological Institute; Location of maximum effect (*p* < 0.05, family-wise error corrected with alphasim, cluster > 215) was described in the MNI space. * Cluster (*p* < 0.0045, correlation with ASMI, after controlling for age, gender and LEDD); ** The FA, MD, AD, and RD values were compared between the PDNSa and PDSa by analysis of covariance after controlling of age, gender and LEDD.

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
