# Peer review of "Correlation between Executive Network Integrity and Sarcopenia in Patients with Parkinson’s Disease"

_ijerph, 2019, doi:10.3390/ijerph16244884_

Round 1
Reviewer 1 Report
Thank you for the opportunity to review this well-written manuscript. I have detected some minor grammatical errors but apart from that, I don't have any other comments to share as I find that the manuscript is well-written and discussion points, as well as the conclusion, have all been well-thought.
Minor comments:
Please have a final read-through manuscript as there are some minor grammatical errors present in the current version e.g. line 67
'The 52 PD patients included in the study (21 men and 31 women; mean age 61.7±10.6 years) were without a history of other neurological or psychiatric illnesses...' would read better as 'The 52 PD patients included in the study (21 men and 31 women; mean age 61.7±10.6 years) had no history of other neurological or psychiatric illnesses...
Table 1, 3rd row: not clear what the following values are related to
10(52.6%) 21(40.4%) 0.357
Author Response
Thank you for the comments and suggestions.
I've already deleted the table 1, 3rd row. It isn't necessary for the table.
Also, I checked the grammatical errors in the manuscript. Please see the new version of manuscript.
Reviewer 2 Report
General Comments
General Weaknesses
- Manuscript presents too many abbreviations at the moment.
- The Authors must improve Introduction. Rationale is weak at the moment (the Authors do not justify why their study is important for literature). This is important weakness of the manuscript (lines 42-64, page 2 of 11).
- Did the Authors verify the whole conditions to use the multivariate analysis of covariance? (lines 125-127, page 3 of 11).
General Strengths
- This section is (Results) very interesting. It contents abundant useful data for readers and researchers (pages 4-7 of 11). This is one of the most important strengths of the manuscript.
- Discussion is very interesting. The Authors discuss the study´s outcomes with other outcomes of similar studies and they hypothetise the possible underlying mechanism of their study´s outcomes (considering enough, pertinent and updated quotations). This is one of the most important strengths of the manuscript (pages 11-17 of 11).
- Tables 1, 2 and 3 and Figures 1 and 2 are correct and clear and they present a lot of interesting and useful data.
Major Comments:
Title:
Weaknesses
- Title must elaborate again. It is “too ambitious” at the moment considering that the study´s limitations are significant (page 1 of 11).
Abstract
Weaknesses
- Abstract must be elaborated again. It must follow recommendations of the present report (page 1 of 11).
Keywords
Weaknesses
- Keywords must be corrected. Authors must avoid use the same words in Title and Keywords (page 1 of 11).
Introduction
Weaknesses
- The Authors must improve Introduction. Rationale is weak at the moment (the Authors do not justify why their study is important for literature). This is important weakness of the manuscript (lines 42-64, page 2 of 11).
Methods
Weaknesses
- Did the Authors get approval from some ethics committee to carry out the research project? (lines 66-79, page 2 of 11 -Participants-).
- Did the Authors register this manuscript as a clinical trial (clinicaltrials.gov, clinicaltrialsregister.eu, etc.) (lines 66-79, page 2 of 11 -Participatns-).
- Did the Authors follow the CONSORT guidelines to elaborate on this manuscript? (lines 66-79, page 2 of 11 -Participants-).
- Did the Authors get the inform consent from participants to participate in the research project? (lines 66-79, page 2 of 11 -Participants-).
- Did the Authors verify the whole conditions to use the multivariate analysis of covariance? (lines 125-127, page 3 of 11).
- Methods must follow the CONSORT guidelines (and the rest of the manuscript as well).
Results
Weaknesses
- Did the Authors elaborate a flow´s chart to present the study´s algorithm? (page 4 of 11).
Strengths
- This section is (Results) very interesting. It contents abundant useful data for readers and researchers (pages 4-7 of 11). This is one of the most important strengths of the manuscript.
Discussion
Strengths
Discussion is very interesting. The Authors discuss the study´s outcomes with other outcomes of similar studies and they hypothesise about the possible underlying mechanism of their study´s outcomes (considering enough, pertinent and updated quotations). This is one of the most important strengths of the manuscript (pages 11-17 of 11).
Conclusions
Weaknesses
- Conclusions are interesting but they must be carefully considered since the study´s limitations are significant (page 9 of 11).
References
- This section must be checked it in detail. It could contain format mistakes (pages 10-11 of 11).
Tables and Figures
Strengths
- Tables 1, 2 and 3 and Figures 1 and 2 are correct and clear and they present a lot of interesting and useful data.
Round 2
Reviewer 2 Report
General Comments General Weaknesses
- Manuscript has been improved, nevertheless, Statistical Analysis must be improved. The Authors do not make clear if they have verified conditions to apply the whole statistical tests that they have used. Considering Statistical Analysis section, I recommend to be clear and simple.
Author Response
The comments and suggestions are much appreciated. The "Analysis of body composition differences between groups" (line 113-114, page3/13) and "Analysis of regions of interest"(line 113-114, page3/13) have been altered accordingly. We hope that the changes have improved the quality and readability of the manuscript.